# Nutritional Composition of Brazilian Food Products Marketed to Children

**DOI:** 10.3390/nu11061214

**Published:** 2019-05-28

**Authors:** Martha Luisa Machado, Vanessa Mello Rodrigues, Amanda Bagolin do Nascimento, Moira Dean, Giovanna Medeiros Rataichesck Fiates

**Affiliations:** 1Nutrition in Foodservice Research Centre, Nutrition Post Graduation Program, Federal University of Santa Catarina, Florianópolis 88040-900, Brazil; marthalmachado@gmail.com (M.L.M.); v.mellorodrigues@yahoo.com.br (V.M.R.); 2Nutrition in Foodservice Research Centre, Department of Nutrition, Federal University of Santa Catarina, Florianópolis 88040-900, Brazil; amanda.bagolin@ufsc.br; 3Institute for Global Food Security, School of Biological Sciences, Queen’s University Belfast, Belfast BT7 1NN, UK; moira.dean@qub.ac.uk

**Keywords:** children, marketing, nutrition labeling, food products, packages, nutrient content

## Abstract

Most food industry marketing in products targeted at children is found in packages of foods containing either excessive fat, sugar, or salt. This study audited all 5620 packaged foods available in a store of a large Brazilian supermarket chain and retrieved information from the nutrition facts tables on package labels. Products were photographed for further visual analysis to determine the presence of marketing strategies directed at children. Comparison of nutrient content per 100 g between children’s and non-children’s food products employed the Student t-test or the Mann–Whitney U-test (*p*-value < 0.05), due to the non-normal distribution of the nutritional composition data as verified through the Shapiro–Wilk test. Brazilian children’s food products from groups 4, 5, and 7 presented higher carbohydrate content than similar non-children’s products, while children’s food products from groups 1 and 7 presented lower fiber content. Results indicate that regulation on food labeling needs revising as it has not been effective in stopping the marketing of energy-dense nutrient-poor foods towards children.

## 1. Introduction

Before being made available to consumers, food products must be appropriately packaged to preserve the food and protect it against contamination. Packaging also extends a product’s shelf-life and provides information to consumers [1].

The design, shape, color, materials, presence of characters, drawings, games, and association with free gifts are used in packaging as communication tools to draw people’s attention to products, including children [2,3,4]. Such commercial communication, designed to increase either the recognition, appeal, or consumption of particular products, is categorized as marketing by the World Health Organization (WHO) [3,5].

The marketing of food and beverages to children influences their preferences and can lead to increased consumption, as children ask for products and influence parents’ purchases. There is evidence that most food industry marketing that conveys fun and fantasy in products targeted at children is found in packages of foods containing either excessive fat, sugar, or salt. Consumption of such foods has been increasing over the last decade and has been associated with a high incidence of overweight, obesity, and non-communicable diseases in children [5,6,7,8,9,10,11,12,13,14,15].

Research conducted in the United States and in the United Kingdom (UK), which compared the nutritional composition of breakfast cereals, yogurt, cereal bars, and ready meals, demonstrated that products targeted to children contained higher amounts of either energy, sodium, sugar, or fat than similar foods that were not targeted to them [16,17]. The North American study found that children’s breakfast cereals contained more energy, sugar, and sodium, and less fiber and protein than those breakfast cereals not marketed to them. It also identified that most (66%) children’s cereals failed to meet national nutrition standards, particularly with respect to sugar content [16]. The UK study concluded that children’s yogurts and cereal bars contained higher amounts of total sugars, fat, and saturated fat than their non-children counterparts [17].

The present study aimed to widen the range of evaluated food products, gathering data on the nutritional composition of the whole range of packaged foods available in a large Brazilian supermarket store, in order to compare the nutritional quality between children’s and non-children’s products. This is particularly relevant considering that Brazil, similarly to other developing economies, has experienced an increasing trend towards the purchase and consumption of processed food products [15,18]. Also, overweight and obesity rates among Brazilians are higher than global averages, ranking fifth among all countries in the number of obese people [19]. In 2009, the prevalence of overweight in 6–11-year-olds was 34% among boys, and 30% among girls, whilst obesity rates among 5–9-year-olds reached 15% [20].

## 2. Materials and Methods

### 2.1. Study Design

This cross-sectional study was undertaken with a dataset of food composition information retrieved in-store from labels of all packaged food items available in a large store pertaining to a major Brazilian supermarket chain. The store belongs to one of the ten largest Brazilian chains in market share (according to the Brazilian Supermarket Association ranking), with 27 stores throughout the country. Therefore, the majority of available products were from big name brands, similar to those sold in other supermarket chains in Brazil. Considering the supermarket had several stores in the city, we chose the one with the highest number of retail products.

Information on energy, carbohydrate, protein, total fat, saturated fat, fiber, sodium content per serving, and serving size was gathered and subsequently standardized for 100 g portions. All food packages were photographed with smartphone cameras to allow identification of marketing strategies towards children. The data collection procedures and dataset preparation are described, in detail, elsewhere [21].

### 2.2. Definition of Children’s and Non-Children’s Food Products

Data from 5620 food products were obtained in the audit. Photos from all food packages were analyzed for the presence of marketing strategies by two researchers. A review of publications on the subject was conducted to inform the definition of what constituted a marketing strategy towards children [3,16,17,22]. A food product was classified as a children’s product if at least one of the following marketing strategies was identified on the front-of-package label: words and phrases such as “child” or “ideal snack for your child”; cartoons, TV series, or film characters; own-brand characters; child celebrities; images of creatures; games or hobbies; colors or shapes that appeal to children; or free gifts. Packages without any of the mentioned marketing strategies were considered as pertaining to a non-children’s product.

### 2.3. Categorization of Products into Food Groups

All products were categorized into one of the eight groups established by Brazilian and Mercosur regulations on labeling [23,24] that classify foods according to their main source of energy: (1) baking goods, breads, cereals, legumes, roots, tubers, and related products; (2) fresh and canned vegetables; (3) fruits, juices, syrups, and drink mixes; (4) milk and dairy products; (5) meat and eggs; (6) oils, fats, and nuts; (7) sugars and products in which carbohydrates and fats are the main energy sources; and (8) gravies, sauces, ready-made seasonings, broths, soups, and ready-to-eat dishes.

During the categorization process, non-children’s products which did not have a children’s product counterpart (*n* = 1457) were identified and excluded from the comparative statistical analysis. These included products such as oatmeal, cereal bars, flours in general, beans, soy-based products, milk, canned tuna, canned sardines, deli meats, sauces, dried fruits, jellies, and seasonings.

### 2.4. Statistical Analysis

Descriptive variables relating to the total number of children’s and non-children’s packaged food products, overall and per food group, are presented by absolute and relative frequencies. Comparison between the total number of children’s and non-children’s products by food group employed the Pearson’s chi-square test (Table 1).

After exclusion of the non-children’s products without a children’s product counterpart, a statistical comparative analysis was performed. Comparison of energy, carbohydrates, total and saturated fat, proteins, fiber, and sodium content between children’s and non-children’s food products employed the Student t-test or the Mann–Whitney U-test (*p*-value < 0.05) (Shapiro–Wilk test determined the normality of distribution of the nutritional composition data). Due to the small number of products targeted at children in groups 2 and 3, a comparison of composition between the children’s and non-children’s products for these groups was not possible. Stata version 11.0 (Stata Corp, College Station, TX, USA) was employed in the analyses. Previous analyses performed on this database, aimed at assessing the nutritional quality of children’s food products, are described elsewhere [21].

## 3. Results

Marketing strategies directed at children were identified in 535 (9.5%) from the 5620 audited packaged foods available in the supermarket store. The remaining 5085 were considered non-children’s food products. Table 2 presents examples of food products categorized in each group established by Brazilian and Mercosur Regulations on Labeling.

Distribution of children’s and non-children’s food products (*n* = 5620) in groups according to the Brazilian and Mercosur labeling regulation and the proportion of children’s food products in each group (*p* < 0.05) are presented in Table 1. Group 7 (sugars and products where carbohydrates and fats are the main energy sources) presented the highest amount of food items (33.7%), and more than half (56.1%) of the 535 children’s food products belonged to this group. It is worth noting that a very small number of products marketed at children were present in groups 2 and 3. Fruits and vegetables which were not fresh but preserved in syrup, sugar, or brine, such as candied fruits, fruits in syrup, and canned vegetables, were classified in groups 7 and 8, due to their nutritional characteristics. Among the beverages, group 3 did not include the soy-based beverages with fruit juice—these were classified in groups 7 and 8 due to their nutritional characteristics. Results of Pearson’s chi-square test showed that group 7 was the only one in which the percentage was significantly higher for children’s products compared to non-children’s products. Groups 1, 2, 3, 6, and 8 presented significantly higher percentages of non-children’s than children’s products.

Table 3 and Table 4 present the statistical comparison by food groups considering only equivalent food products (535 children’s, 3628 non-children’s). In group 1, children’s products presented lower fiber content than non-children’s products. In group 4, children’s products were lower in energy, total fat, saturated fat, protein, and sodium, but higher in carbohydrates than non-children’s products. In group 5, children’s products contained less total fats, saturated fats, and sodium, and more fiber than non-children’s products, but had more carbohydrates. In group 6, children’s products had higher saturated fat and protein content and less sodium content. In group 7, children’s products were higher in carbohydrate and lower in total and saturated fats and fiber. In group 8, children’s products presented less carbohydrates and sodium than non-children’s products. All reported differences were statistically significant (*p* < 0.05).

## 4. Discussion

More than half of the 535 children’s food products belonged to group 7 (sugars and products in which carbohydrates and fats are the main energy sources), and the comparison revealed that this group was the only one in which the percentage was significantly higher for children’s products compared to non-children’s products. Children’s products from groups 4 (milk and dairy products), 5 (meats and eggs), and 7 presented higher carbohydrate content per 100 g than similar non-children’s products, while children’s food products from groups 1 (baking goods, breads, cereals, legumes, roots, tubers) and 7 presented lower fiber content in comparison with non-children’s products.

Products targeted at children from group 1 were mainly breakfast cereals, bread, and crackers manufactured with refined flours, while those not marketed to children in the same group contained more wholegrain ingredients (granola, cereal bars, wholegrain pasta, and wholegrain crackers). The same difference in ingredient sources was observed in children and non-children’s products from groups 4 (milk and dairy products) and 7 (sugars and products where carbohydrates and fats are the main energy sources), indicating that the use of refined ingredients is higher in the manufacture of products that bear marketing strategies to children. Additionally, this might explain the lower fiber content identified when comparing children and non-children’s products from group 1, and the higher amounts of carbohydrate found in groups 4 and 7. This is relevant if we consider that recently, the Global Burden of Disease Study estimated that, in 2017, the low intake of whole grains was globally responsible for 3 million deaths and 82 million disability-adjusted life-years [25].

The lowest amounts of children’s food products were identified in groups 2 and 3 (vegetables and fruits, respectively). The results are in alignment with findings from a Canadian study showing that only 1% of foods targeted at children were represented by fruits and vegetables [22]. Additionally, our study results are in accordance with what was reported in the United States [26], Guatemala [27], Belgium [28], and Australia [29], showing that the food and drink marketing industry aimed at children mainly promotes soft drinks, savory snacks, confectionery, and cookies.

In 2011, the Pan American Health Organization (PAHO) published the report “Recommendations from a Pan American Health Organization Expert Consultation on the Marketing of Food and Non-Alcoholic Beverages to Children in the Americas”. According to this report, packaging as a marketing strategy directed exclusively to children and with specific appeal to children must be prohibited. The document states that only “whole foods” are part of a healthy diet and can be marketed to children without restriction. Whole foods would be those belonging to the following food groups, with no added sweeteners, sugar, salt, or fat: fruits, vegetables, whole grains, fat-free or low-fat dairy products, fish, meat, poultry, eggs, nuts and seeds, and beans. In the case of beverages, the recommendation is clean potable water [5]. In our study, children’s dairy products (group 4) presented significantly higher levels of carbohydrate than non-children’s products. Similar results were found by Lythgoe et al. (2013) for yogurts targeted at children, which contained significantly higher levels of energy, carbohydrates, sugars, total fat, and saturated fat per 100 g when compared with non-children’s products [17]. Our results also revealed that children’s food products from group 1 (which included rice, pasta, pre-fried/frozen tubers, biscuits, breakfast cereals, granola, rice flour, sliced bread, bread rolls, custard powder mix, and cake mix) contained lower fiber content than non-children’s products. In the United States, Schwartz et al. (2008) found similar results in the comparison between non-children and children’s cereals, which were less dense in fiber and protein, but denser in energy, sugar, and sodium [16].

Notwithstanding the higher amounts of carbohydrates in children’s products from groups 4, 5 and 7, the Brazilian regulation for nutrition labeling states that disclosing data for total sugars content is not mandatory [30]. Besides that, previous analysis performed on this database aimed at assessing the presence and types of added sugars in packaged foods identified that most of these contain added sugars, which may hamper adherence to the recommendation of limiting added sugar intake [31].

One of the study’s limitations is that comparison was undertaken between food groups and not individual items, therefore results may not correspond to the individual characteristics of a particular food product. Also, the aim was not to determine if levels were excessive, and therefore it is possible that both categories of products contained similarly high amounts of nutrients, for example, salt and fats. Nevertheless, considering that the latter are already the target of many strategies directed towards their reduction, it is possible that the food industry is already complying and managing to lower the levels of such ingredients in processed foods [32,33].

This study contributes to informing Brazilian policymakers on how current regulation on the use of marketing strategies towards children on packages of food products has not been effective at stopping the marketing of energy-dense nutrient-poor foods [34] towards children. In the United Kingdom, France, Spain, Canada, United States, and Mexico, strategies have already been implemented by authorities to regulate food marketing on children’s packaged foods [5,35,36,37], yet there is not a global recommendation on this subject. In 2011, the PAHO published recommendations from a panel of experts on the promotion and publicity of food and drinks to children in the Americas [5], encouraging the use of marketing strategies only on those foods considered part of a healthy diet; however, in Brazil, self-regulation is still in place, and manufacturers themselves define which foods will bear marketing strategies on their labels.

The comparison between children’s and non-children’s food products revealed that children’s products do not have better nutritional quality than non-children’s food products and, therefore, they should not be allowed to bear marketing strategies to children. As our results revealed that the groups with the highest number of products were from groups 1 and 7, both containing carbohydrate-rich foods, we recommend the definition and implementation of cutoff points by regulatory agencies to guide the permission to use marketing strategies in products directed to children. For this to be possible, it is necessary that disclosure of the quantity of added sugar be made mandatory, as part of Brazil’s food labeling regulation.

## Figures and Tables

**Table 1 nutrients-11-01214-t001:** Distribution of packaged food products available in a large Brazilian supermarket store and proportion of children’s food products into the food groups established by the Brazilian/Mercosur Regulation (*n* = 5620).

Group	Total of Food Products % (*n*)	Children’s Food Products % (*n*)	Non-Children’s Food Products % (*n*)	% Distribution of Children’s Food Products/Total of Children’s Food Products (*n* = 535)	% Distribution of Non-Children’s Food Products/Total of Non-Children’s Food Products (*n* = 5085)	*p*
G1—Baking goods, breads, cereals, legumes, roots, tubers	19.6 (1102)	6.5 (72)	93.5 (1030)	13.5	20.3	<0.001 *
G2—Fresh and canned vegetables	8.3 (464)	1.7 (8)	98.3 (456)	1.5	9.0	<0.001 *
G3—Fruits, juices, syrups, and drink mix	4.8 (271)	3.0 (8)	97.0 (263)	1.5	5.2	<0.001 *
G4—Milk and dairy products	10.9 (614)	10.3 (63)	89.7 (551)	11.8	10.8	0.507
G5—Meats and eggs	4.8 (269)	8.6 (23)	91.4 (246)	4.3	4.8	0.579
G6—Oils, fats, and nuts	6.8 (383)	5.2 (20)	94.8 (363)	3.6	7.1	0.003 *
G7—Sugars and products in which carbohydrates and fats are the main energy sources	33.7 (1895)	15.8 (300)	84.2 (1595)	56.1	31.4	<0.001 *
G8—Gravies, sauces, ready-made seasonings, broths, and ready-to-eat dishes	11.1 (622)	6.6 (41)	93.4 (581)	7.7	11.4	0.008 *
**Total**	**100.0 (5620)**	**9.5 (535)**	**90.5 (5085)**	**100.0**	**100.0**	

* Statistically significant at *p* < 0.05 (*Pearson’s chi-square test*).

**Table 2 nutrients-11-01214-t002:** Examples of food products available in the supermarket, classified in the eight groups established by Brazilian and Mercosur Regulation.

Food Groups	Examples of Food Products
1. Baking goods, breads, cereals, legumes, roots, tubers	Rice, pasta, pre-fried/frozen tubers, biscuits, breakfast cereals, granola, rice flour, sliced bread, bread rolls, custard powder mix, cake mix
2. Fresh and canned vegetables	Tomato sauce, sweet corn, cherry tomatoes, baby carrots
3. Fruits, juices, syrups, and drink mixes	Fruit juices, fruit nectars, apples
4. Milk and dairy products	Milk-based beverages, fermented milk, yogurt, puddings, pudding powder mix, petit-Suisse cheese
5. Meats and eggs	Meatballs, burgers, sausages, bologna, eggs, chicken nuggets, fish fingers
6. Oils, fats, and nuts	Bacon, spreads, coconut milk, peanuts, grated coconut
7. Sugars and products in which carbohydrates and fats are the main energy sources	Sweet spreads, honey, syrups, gelatin powder mix, candies, gum, chocolate, ice-cream, crisps, condensed milk, cakes, ready-to-eat popcorn
8. Gravies, sauces, ready-made seasonings, broths, and ready-to-eat dishes	Ketchup, lasagna, savory pie, pizza, noodles

**Table 3 nutrients-11-01214-t003:** Comparison of energy, and total and saturated fat content per 100 g of children’s (C) (*n* = 535) and non-children’s (NC) (*n* = 3628) food products.*

		Energy (kcal)				Total Fat (g)				Saturated Fat (g)		
Category	*n*	Median	IQR	*p*	*n*	Median	IQR	*p*	*n*	Median	IQR	*p*
**Group 1**												
**NC**	627	353.7	340.0–393.3	0.159	609	2.8	1.2–10.3	0.141	597	0.6	0.0–2.7	0.481
**C**	72	356.0	307.5–370.7		68	2.2	1.0–7.0		68	0.6	0.0–1.6	
**Group 4**												<0.001 ^a^
**NC**	516	149.5	78.7–300.0	<0.001 ^a^	482	5.1	1.7–23.0	<0.001 ^a^	478	2.7	1.1–13.3	
**C**	63	82.3	73.7–115.0		56	2.3	1.5–3.0		56	1.5	0.8–2.0	
**Group 5**												
**NC**	178	210.5	167.3–248.3	0.267	178	13.9	10.0–18.0	0.007 ^a^	176	4.8	3.0–6.8	0.006 ^a^
**C**	23	204.0	170.0–216.1		23	10.0	8.5–15.0		23	3.3	2.2–4.2	
**Group 6**												
**NC**	149	560.0	460.0–626.7	0.383	149	45.3	31.0–60.0	0.702	149	9.3	6.7–18.0	<0.001 ^a^
**C**	20	558.3	253.3–660.0		20	45.0	20.0–72.5		20	20.0	10.2–41.7	
**Group 7**												
**NC**	1371	400.0	262.6–506.7	0.364	1026	19.6	8.3–29.3	<0.001 ^a^	1004	7.6	3.1–14.4	<0.001 ^a^
**C**	300	392.7	348.2–470.0		214	16.0	0.8–22.0		205	5.5	0.0–9.0	
**Group 8**												
**NC**	357	232.5	123.0–372.0	0.515	332	8.0	3.9–16.5	0.251	327	3.1	1.0–6.6	0.068
**C**	41	218.2	128.0–275.5		40	8.8	5.8–15.6		40	3.6	2.7–6.2	

* Non-children’s products which did not have a children’s product counterpart (*n* = 1457) were excluded from this statistical comparative analysis. Due to the small number of children’s food products in groups 2 and 3 (*n* = 16), comparison was not possible. IQR: interquartile range; *n*: number; NC: non-children; C: children; Group 1: baking goods, breads, cereals, legumes, roots, tubers. Group 4: milk and dairy products. Group 5: meats and eggs. Group 6, oils, fats, and nuts. Group 7: sugars and products in which carbohydrates and fats are the main energy sources. Group 8: gravies, sauces, ready-made seasonings, broths, and ready-to-eat dishes. ^a^ Statistically significant at *p* < 0.05 (Mann–Whitney U-test).

**Table 4 nutrients-11-01214-t004:** Comparison of protein, carbohydrate, fiber, and sodium content per 100 g children’s (C) (*n* = 535) and non-children’s (NC) (*n* = 3628) food products.*

		Protein (g)				Carbohydrates (g)				Fiber (g)				Sodium (mg)		
Categories	*n*	Median	IQR	*p*	*n*	Median	IQR	*p*	*n*	Median	IQR	*p*	*n*	Median	IQR	*p*
**Group 1**																
**NC**	620	9.4	6.7–11.5	0.097	627	70.0	60.0–76.2	0.476	611	3.0	2.0–6.0	0.002 ^a^	619	106.7	0.0–391.9	0.144
**C**	70	7.6	6.0–11.0		72	70.6	52.5–80.0		70	2.4	0.6–4.4		72	290.0	0.0–449.8	
**Group 4**																
**NC**	496	4.6	2.9–13.3	<0.001 ^a^	511	9.3	3.0–15.5	<0.001 ^a^	394	0	0.0–0.0	0.509	514	83.3	46.9–468.0	<0.001 ^a^
**C**	60	2.3	2.1–6.0		63	14.4	12.8–16.2		42	0	0.0–0.0		63	44.4	36.0–71.5	
**Group 5**																
**NC**	178	12.5	10.0–16.0	0.410	178	3.0	0.6–7.2	<0.001 ^a^	171	0	0.0–0.0	<0.001 ^a^	177	720.0	454.5–1089.5	0.011 ^a^
**C**	23	12.3	12.0–13.1		23	12.5	3.2–19.2		22	0.9	0.0–1.7		23	516.1	346.1–747.5	
**Group 6**																
**NC**	130	0	0.0–20.8	0.009 ^a^	129	16.0	0.0–30.4	0.281	131	5.6	0.0–8.7	0.062	140	406.7	35.0–705.0	0.580
**C**	12	5.8	0.0–8.3		12	9.3	0.0–25.8		12	0	0.0–6.7		11	91.6	0.0–650.0	
**Group 7**																
**NC**	1053	5.8	3.5–7.6	0.321	1365	56.7	28.6–70.0	<0.001 ^a^	1009	2.0	0.0–3.2	0.007 ^a^	1263	100.0	30.0–276.7	0.070
**C**	235	5.6	3.6–7.2		300	68.0	54.3–80.0		200	0.9	0.0–2.4		275	120.0	31.0–316.7	
**Group 8**																
**NC**	334	8.2	5.7–10.2	0.601	358	26.3	16.0–56.0	0.008 ^a^	330	2.0	1.0–3.2	0.187	359	629.6	372.1–1233.3	0.026 ^a^
**C**	40	7.8	6.1–10.2		41	22.1	11.7–28.6		40	1.9	1.0–2.5		41	396.7	349.5–648.0	

* Non-children’s products which did not have a children’s product counterpart (*n* = 1457) were excluded from this statistical comparative analysis. Due to the small number of children’s food products in groups 2 and 3 (*n* = 16), a comparison was also not possible. IQR: interquartile range; *n*: number; NC: non-children; C: children. Group 1: baking goods, breads, cereals, legumes, roots, and tubers. Group 4: milk and dairy products. Group 5: meats and eggs. Group 6: oils, fats, and nuts. Group 7: sugars and products in which carbohydrates and fats are the main energy sources. Group 8: gravies, sauces, ready-made seasonings, broths, and ready-to-eat dishes. ^a^ Statistically significant at *p* < 0.05 (Mann–Whitney U-test).

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
