# Peer review of "Nutritional Composition of Brazilian Food Products Marketed to Children"

_nutrients, 2019, doi:10.3390/nu11061214_

Round 1

Reviewer 1 Report

The manuscript entitled “Nutritional composition of Brazilian food products marketed to children“ presents interesting and up to date issues, but it requires also some corrections before being published.

Abstract:

- Lines 22 - “Brazilian products targeted at children ….” Please specify the percentage of such products, or state that “All Brazilian food products targeted at children …..”

Keywords: Authors should add more keywords than three, as they do not represent the conducted study. 

Introduction:

- Line 32 - “pathogenic contamination.” Not only pathogenic contaminations are relevant, but also others, e.g. chemical, physical, etc. Authors could omit the word “pathogenic”

Research Methodology:

- I have great concerns regarding the applied methodology. Some of the food groups dedicated for children are represented by a very small number of products (e.g.; G2 and G3 by 8 products only). On the basis of such a small number of items, conclusions should have been drawn very carefully. Moreover the proportion between children and non-children food products are not optimal. Therefore the table 1 may be misleading. Such small groups (G2 and G3) should be removed and the data should be recalculated. 

- Moreover, it should be indicated if analyzed products have excessive amount of specific nutrients or have only a little bit higher but acceptable level of nutrients (instead of limitations sections). The presented analysis is only relative comparison (which products have lower or higher level of nutrients). Of course the food products for children should be monitored in term of excessive contents of saturated fats or sugar. But the conducted analysis do not present the whole picture. 

Results:

- Lines 128-129 – “In Group 7, children products were higher in carbohydrates and sodium, and lower in total and saturated fats and fiber” In the case of sodium p=0.07, whereas the α=0.05, therefore it is NOT statistically significant difference (for the assumed criteria, they are not higher in indicated nutrients, but they do not differ). 

Discussion section:

- The discussion section should be non-judgmental – Authors should objectively present the situation. However, it was not done. 

- The first sentence “Brazilian products targeted at children presented higher energy and carbohydrate content, and lower fiber content per 100g of product” is not accordance with the findings (Lines 124-125 – “In Group 4, children products were lower in energy, total fat, saturated fat, protein, and sodium, but higher in carbohydrates than non-children products.” Lines 126-127 – “In Group 5, children products presented less total fats, saturated fats, and sodium, and more fiber than non-children products, but more carbohydrates”). 

- Of course, Authors should have emphasized the negative aspects of SOME of the products for children but they should not generalize.  

- Lines 161 “…5-10% of the daily energy amount consumed”. This sentence could be misleading, due to the fact, that WHO recommends “reducing the intake of free sugars to less than 10% of total energy intake” but in term of accuracy it should be calculated for total energy intake, not for reported energy intake

- There is almost no discussion. This section must be extended. Authors should relate the findings to those of similar studies and point the differences and similarities between the studies and their results. Authors should add the appropriate references in this section.

Conclusion:

- The conclusion is not associated directly with the findings. 

Author Response

Authors' reply to the Review Report (Reviewer 1)

Q1. The manuscript entitled “Nutritional composition of Brazilian food products marketed to children” presents interesting and up to date issues, but it requires also some corrections before being published.

A. Thank you for recognizing the manuscript’s potential and giving us the opportunity to improve it.

Q2. Abstract - Lines 22 - “Brazilian products targeted at children ….” Please specify the percentage of such products, or state that “All Brazilian food products targeted at children …..”

A. The sentence starting at line 22 was rewritten and now reads: Brazilian products targeted at children from groups 4, 5 and 7 presented higher carbohydrate content, while products from groups 1 and 7 presented and lower fiber content per 100g of product than similar products not targeted at them.

Q3. Abstract - Keywords: Authors should add more keywords than three, as they do not represent the conducted study.

A. Three more keywords were added: Food products; Packages; Nutrient content

Q4. Introduction - Line 32 - “pathogenic contamination.” Not only pathogenic contaminations are relevant, but also others, e.g. chemical, physical, etc. Authors could omit the word “pathogenic”

A.Thank you for pointing this out. The word “pathogenic” was removed from the sentence.

Q5. Research Methodology - I have great concerns regarding the applied methodology. Some of the food groups dedicated for children are represented by a very small number of products (e.g.; G2 and G3 by 8 products only). On the basis of such a small number of items, conclusions should have been drawn very carefully. Moreover the proportion between children and non-children food products are not optimal. Therefore the table 1 may be misleading. Such small groups (G2 and G3) should be removed and the data should be recalculated.

A.This is absolutely right, this is why groups 2 and 3 only appear in Table 1, which is descriptive. Table 2, presenting comparison between groups, does not contain data from either group. In order to make this more clear, a sentence was added in the methods section (page3, lines 108-110): Due to the small quantity of products targeted at children in Groups 2 and 3, comparison between children and non-children products’ composition was not possible).

Q6. Research Methodology - Moreover, it should be indicated if analyzed products have excessive amount of specific nutrients or have only a little bit higher but acceptable level of nutrients (instead of limitations sections). The presented analysis is only relative comparison (which products have lower or higher level of nutrients). Of course the food products for children should be monitored in term of excessive contents of saturated fats or sugar. But the conducted analysis do not present the whole picture.

A.The reviewer is right. But, as now declared in page 3, lines 111-112, previous analyses performed on this database aimed to assess the nutritional quality of children food products are described elsewhere (Rodrigues et al., 2017).

Q7. Results - Lines 128-129 – “In Group 7, children products were higher in carbohydrates and sodium, and lower in total and saturated fats and fiber” In the case of sodium p=0.07, whereas the α=0.05, therefore it is NOT statistically significant difference (for the assumed criteria, they are not higher in indicated nutrients, but they do not differ).

A.This was mistakenly reported. Page 4, lines 140-141 now reads: In Group 7, children products were higher in carbohydrate and lower in total and saturated fats and fibre.

Q8. Discussion section - The discussion section should be non-judgmental – Authors should objectively present the situation. However, it was not done.

A.We are sorry for this, and thank the reviewer for the comment. The discussion section was entirely re-written. Hopefully now it is more argumentative and less judgmental.

Q9. Discussion section - The first sentence “Brazilian products targeted at children presented higher energy and carbohydrate content, and lower fiber content per 100g of product” is not accordance with the findings (Lines 124-125 – “In Group 4, children products were lower in energy, total fat, saturated fat, protein, and sodium, but higher in carbohydrates than non-children products.” Lines 126-127 – “In Group 5, children products presented less total fats, saturated fats, and sodium, and more fiber than non-children products, but more carbohydrates”). Of course, authors should have emphasized the negative aspects of SOME of the products for children but they should not generalize. 

A. The first sentence was removed from the discussion section.  The discussion section was entirely re-written in order not to generalize and focus more specifically in differences between certain groups.

Q10. Discussion section - Lines 161 “…5-10% of the daily energy amount consumed”. This sentence could be misleading, due to the fact, that WHO recommends “reducing the intake of free sugars to less than 10% of total energy intake” but in term of accuracy it should be calculated for total energy intake, not for reported energy intake

A. We agree with the reviewer. The sentence was rewritten.

Q12. Discussion section - There is almost no discussion. This section must be extended. Authors should relate the findings to those of similar studies and point the differences and similarities between the studies and their results. Authors should add the appropriate references in this section.

A. Discussion section was extended and improved. Findings are now related to those of similar studies, as well as differences and similarities. Appropriate references were added.

Q13. Conclusion - The conclusion is not associated directly with the findings.

A. The conclusion was also modified, hopefully now it is more in line with what the reviewer judges as appropriate (please see page 9, lines 227-234).

Reviewer 2 Report

This is a useful study, providing data on marketing of foods to children in Brazil. This is important topic in public Health, because children are a vulnerable target population. While authors checked if foods targeting children differ from others in respect to certain Nutrients, it would be very useful if overall Food quality could be taken into account. Considering that WHO has made a considerable progress and published nutrient profile models specifically for limiting marketing of foods to children, such a model could be used fo this purpose. Here are some other minor issues, that needs to be resolved:

Line 93 – 94: please make clearer, which products were excluded and why. 

Line 146: explain with which products did you make this comparison (to products that are not targeted to children?) 

Line 156 - 159: it seems that in current regulation, information on total amounts of sugar is not mandatory but in line 158 you state that (only?) the amounts of ADDED sugars and other sugary ingredients should be included and made mandatory. Please explain this more thoroughly 

Line 171 – 173: the statement about regulation is a bit superficial, as it states only a few countries and not all of them have the same regulations nor all of them are statutory. Please rewrite this sentence to make it clearer or write it more generally.

Author Response

Authors’ Reply to the Review Report (Reviewer 2)

Q1. This is a useful study, providing data on marketing of foods to children in Brazil. This is important topic in public Health, because children are a vulnerable target population. While authors checked if foods targeting children differ from others in respect to certain Nutrients, it would be very useful if overall Food quality could be taken into account. Considering that WHO has made a considerable progress and published nutrient profile models specifically for limiting marketing of foods to children, such a model could be used for this purpose.

A. We thank the reviewer for the comment. The analyses regarding overall quality of the products using nutrient profile models were conducted previously by the research team and described elsewhere (Rodrigues et al., 2017).

Q2. Line 93 – 94: please make clearer, which products were excluded and why.

A. The paragraph now reads:  “During the categorization process, non-children products which did not have a children product counterpart (n=1,457) were identified and excluded of the statistical comparative analysis. These included products such as oatmeal, cereal bars, flours in general, beans, soy-based products, milk, canned tuna, canned sardines, deli meats, sauces, dried fruits, jellies, and seasonings.”

Q3. Line 146: explain with which products did you make this comparison (to products that are not targeted to children?)

A.Yes, similar products not targeted to children. Please see page, 3, lines 104-112.

Q4. Line 156 - 159: it seems that in current regulation, information on total amounts of sugar is not mandatory but in line 158 you state that (only?) the amounts of ADDED sugars and other sugary ingredients should be included and made mandatory. Please explain this more thoroughly

A. The discussion section was rewritten, so these statements were modified.

Q5.Line 171 – 173: the statement about regulation is a bit superficial, as it states only a few countries and not all of them have the same regulations nor all of them are statutory. Please rewrite this sentence to make it clearer or write it more generally.

A. The discussion section was rewritten, so these statements were modified.

Round 2

Reviewer 1 Report

Authors made great effort to improve the manuscript. I have only one, minor comment:

-        References should be corrected (see author guidelines)

Reviewer 2 Report

Thank you for explanation and addressing the comments.